# Tripartite Interactions among *Ixodiphagus hookeri*, *Ixodes ricinus* and Deer: Differential Interference with Transmission Cycles of Tick-Borne Pathogens

**DOI:** 10.3390/pathogens9050339

**Published:** 2020-04-30

**Authors:** Aleksandra I. Krawczyk, Julian W. Bakker, Constantianus J. M. Koenraadt, Manoj Fonville, Katsuhisa Takumi, Hein Sprong, Samiye Demir

**Affiliations:** 1Centre for Infectious Disease Control, National Institute for Public Health and the Environment, Antonie van Leeuwenhoeklaan 9, 3721 MA Bilthoven, The Netherlands; aleksandra.i.krawczyk@gmail.com (A.I.K.); manoj.fonville@rivm.nl (M.F.); katsuhisa.takumi@rivm.nl (K.T.); 2Laboratory of Entomology, Wageningen University & Research, 6708PB Wageningen, The Netherlands; julian.bakker@wur.nl (J.W.B.); sander.koenraadt@wur.nl (C.J.M.K.); 3Zoology Section, Department of Biology, Ege University Faculty of Science, Bornova Izmir 35040, Turkey

**Keywords:** parasitic wasp, biological control, tick-borne pathogen, host preference, parasitization, transmission cycle, Lyme borreliosis, human granulocytic anaplasmosis, neoehrlichiosis

## Abstract

For the development of sustainable control of tick-borne diseases, insight is needed in biological factors that affect tick populations. Here, the ecological interactions among *Ixodiphagus hookeri*, *Ixodes ricinus*, and two vertebrate species groups were investigated in relation to their effects on tick-borne disease risk. In 1129 questing ticks, *I. hookeri* DNA was detected more often in *I. ricinus* nymphs (4.4%) than in larvae (0.5%) and not in adults. Therefore, we determined the infestation rate of *I. hookeri* in nymphs from 19 forest sites, where vertebrate, tick, and tick-borne pathogen communities had been previously quantified. We found higher than expected co-occurrence rates of *I. hookeri* with deer-associated *Anaplasma phagocytophilum*, and lower than expected rates with rodent-associated *Borrelia afzelii* and *Neoehrlichia mikurensis*. The prevalence of *I. hookeri* in nymphs varied between 0% and 16% and was positively correlated with the encounter probability of ungulates and the densities of all life stages of *I. ricinus*. Lastly, we investigated the emergence of *I. hookeri* from artificially fed, field-collected nymphs. Adult wasps emerged from seven of the 172 fed nymphs. From these observations, we inferred that *I. hookeri* is parasitizing *I. ricinus* larvae that are feeding on deer, rather than on rodents or in the vegetation. Since *I. hookeri* populations depend on deer abundance, the main propagation host of *I. ricinus*, these wasps have no apparent effect on tick populations. The presence of *I. hookeri* may directly interfere with the transmission cycle of *A. phagocytophilum*, but not with that of *B. afzelii* or *N. mikurensis*.

## 1. Introduction

Lyme borreliosis poses serious health concerns in the northern hemisphere [1,2,3]. Increases in incidence have been observed in several countries in Europe. These increases are partially explained by geographical spread of its vector, *Ixodes ricinus* [4,5]. Other tick-borne diseases (TBDs), such as anaplasmosis and neoehrlichiosis, are also emerging [6,7]. Understanding which factors drive the population densities of ticks and the transmission cycles of tick-borne pathogens are important steps in assessing disease risk and formulating possible intervention strategies. In north-western Europe, TBDs are generally caused by a bite of an infected *I. ricinus* nymph [8]. Therefore, the density of infected nymphs (DIN) is an important ecological parameter that contributes to the overall disease risk of TBD [9,10,11].

*Ixodes ricinus* has a three-host life cycle and its survival depends on finding vertebrate hosts for feeding and propagation. *Ixodes ricinus* utilizes a multitude of host species, but these species differ in the number of ticks and in the life stages they feed [12]. In a typical north-western European forest, larvae predominantly feed on rodents, nymphs on rodents and birds, and adult ticks on deer, mostly *Capreolus capreolus* [13,14,15]. The presence of deer generally results in a high abundance of *I. ricinus* [15,16,17]. However, fluctuations in rodent density have also been associated with variations in the density of (infected) nymphs [18,19].

Control of *I. ricinus*-borne diseases primarily consists of the promotion of personal preventive actions, but its effectiveness is questionable [20,21,22]. Biological control of ticks is considered to be an environmentally friendly approach. However, so far, this has not been effectively applied in routine settings [23,24,25,26,27]. Interestingly, the parasitoid wasp, *Ixodiphagus hookeri* (Hymenoptera: *Encyrtidae*), is a natural enemy of *I. ricinus*, and has been a target of interest as a biological control agent [23,27,28].

Although the total lifecycle of *I. hookeri* generally takes one year, the adult life stage has a lifespan of only two to three days [23,29]. It is not well understood how female wasps find *I. ricinus* larvae and nymphs for oviposition [23,27]. We hypothesize that female *I. hookeri* randomly infest *I. ricinus* larvae and nymphs in the vegetation. We test this by calculating the infestation rate of the life stages of questing *I. ricinus*. If *I. hookeri* infests ticks in the vegetation, then we expect to find the wasp in nymphs together with horizontally transmitted tick-borne pathogens more or less randomly. The alternative hypothesis is that female *I. hookeri* infest *I. ricinus* larvae while feeding on a vertebrate host. By calculating the co-occurrence rates of the wasp with strictly horizontally transmitted tick-borne pathogens, we might infer a preference for a vertebrate host on which a female wasp would infest feeding ticks with her eggs.

The embryonic development of *I. hookeri* eggs is triggered by the attachment and engorgement of nymphs feeding on a vertebrate host [23,27]. *Ixodiphagus hookeri* larvae feed on the internal tissue along with the vertebrate blood ingested by nymphs [23,27]. Adult wasps emerge by making a hole in the nymph’s body, and kill the tick before molting [30]. Until now, the emergence of *I. hookeri* has only been observed from fully fed *I. ricinus* nymphs [27,30,31]. Although several studies on the interactions between *I. hookeri* and *I. ricinus* have been conducted [27,29,31,32,33,34,35], we lack estimates about the impact of this wasp on *I. ricinus* population dynamics. We hypothesize that the prevalence of *I. hookeri* in nymphs is driven by the abundance of immature stages of *I. ricinus*, and because of their predatory behavior, an increasing prevalence of wasps is negatively associated with the abundance of *I. ricinus* adults. For this, we determined the infestation rates of *I. hookeri* in questing nymphs, and correlated it with the abundances of the different life stages of *I. ricinus* from a cross-sectional study, in which tick density and vertebrate communities were quantified [13,36]. The latter allowed us to determine the relationship between *I. hookeri* and tick hosts such as ungulates and rodents.

Lastly, to determine whether the molecular detection of the wasp in ticks reflects the presence of viable wasp eggs, and to determine to what extent these infestations kill nymphs, an artificial tick feeding assay was performed.

## 2. Results

### 2.1. Prevalence of I. hookeri and Tick-Borne Pathogens in Different Life Stages of I. ricinus

To determine which life stage of *I. ricinus* was most often infested with *I. hookeri*, larvae (*n* = 367), nymphs (*n* = 684), and adults (*n* = 78) were tested (Table 1). *Ixodiphagus hookeri* DNA was found in two larvae (0.5%), 30 nymphs (4.4%), but not in adult *I. ricinus*. Thus, the highest prevalence of *I. hookeri* was detected in nymphs. The three horizontally transmitted tick-borne pathogens, *Borrelia burgdorferi* s.l., *Anaplasma phagocytophilum*, and *Neoehrlichia mikurensis*, were detected in nymphs and adults (Table 1), but not in larvae, except for one larva, which was positive of *B. burgdorferi* s.l. (Table 1).

### 2.2. Infestations with I. hookeri and Infection with Horizontally Transmitted Tick-Borne Pathogens

The presence of *I. hookeri* DNA was determined in 13,967 nymphs from the cross-sectional study, and which had already been tested for the prevalence of tick-borne pathogens [13,36]. *Ixodiphagus hookeri* was detected in nymphs from 18 of the 19 forest sites examined in the Netherlands, with prevalence varying from 0.1% to 15.9% (Appendix A). The infestation with *I. hookeri* and infection with horizontally transmitted pathogens appeared not to be random (Table 2). The presence of *I. hookeri* DNA in nymphs was positively associated with the deer-associated *A. phagocytophilum,* and negatively associated with the rodent-borne *B. afzelii* and *N. mikurensis* (Table 2 and Appendix A). Apparently, the infestation with *I. hookeri* appears somehow to be associated with the first blood meal of the questing nymphs.

### 2.3. Association of I. hookeri Prevalence in Questing Nymphs with Density of I. ricinus

The relationship between prevalence (%) of *I. hookeri* in questing nymphs and the density of all life stages of *I. ricinus* of the 19 forests sites was investigated with generalized linear models. The prevalence (%) of *I. hookeri* was significantly positively associated with the density of questing *I. ricinus* larvae (DOL; *p* < 0.0001), nymphs (DON; *p* < 0.0001) and adults (DOA; *p* < 0.0001; Figure 1). Equations of all models, Akaike information criterion (AIC) values, and results of the likelihood ratio test are provided in Appendix A.

### 2.4. Association of I. hookeri with Densities of Ungulates and Rodents

Previous analyses of the data from the cross-sectional study showed that the abundance of ungulates was positively associated with the density of the three life stages of *I. ricinus* [13], just like the prevalence of *I. hookeri* in our study (Figure 1). The observed positive association of *I. hookeri* with densities of the different life stages of *I. ricinus* might therefore be caused by the dominant role of deer in propagation of ticks. Indeed, the occurrence of *I. hookeri* in questing nymphs was positively correlated with the encounter probability with ungulates (Figure 2, left panel). No significant association between the encounter probability of rodents and *I. hookeri* occurrence was observed (Figure 2, right panel).

### 2.5. Artificial Blood-Feeding of I. ricinus Nymphs

Lastly, to determine whether the detection of *I. hookeri* in ticks reflects the presence of viable wasp eggs, and to determine to what extent these infestations kill nymphs, an artificial tick feeding assay was performed. A total of 561 nymphs collected in the Amsterdamse Waterleidingduinen were placed on feeding membranes, of which 172 successfully blood-fed and engorged (Table 3). A total of 151 nymphs (88%) successfully molted into the adult stage. In total, 21 engorged nymphs did not molt and died (12%); 14 died without and seven died with the emergence of three to five wasps (Table 3). We detected wasp DNA in seven nymphs the wasps emerged from, as well as in nine of the 14 dead nymphs, and in ten of the 151 that engorged and successfully molted to adults (Table 3). In total, 15% of the nymphs were infested with *I. hookeri*, which is comparable to the infestation rate found previously in the Amsterdamse Waterleidingduinen (13%; Appendix A). Thus, from the nymphs infested with *I. hookeri* (*n* = 26), sixteen (62%) died and ten (38%) survived and successfully molted into an adult tick.

## 3. Discussion

In this study, we investigated the ecological interactions among the parasitoid wasp *I. hookeri*, the tick *I. ricinus*, and two vertebrate species groups (ungulates and rodents), and their combined effect on tick-borne disease risk. Significant differences in the prevalence of *I. hookeri* between questing *I. ricinus* larvae and nymphs (Table 1) were found as well as different associations of wasps with horizontally transmitted tick-borne pathogens in nymphs (Table 2). From these results, we inferred that female *I. hookeri* wasps infest engorging larvae while feeding on a deer, rather than on a rodent or in the vegetation. In addition, the positive association of the *I. hookeri* prevalence with the encounter probability of deer, but not rodents (Figure 2) supports the idea that female wasps are attracted to deer by their odor, and subsequently infest ticks that are feeding on these deer [38,39]. Since deer act as major propagation hosts for *I. ricinus*, we observed a positive association between *I. hookeri* prevalence and the density of all tick life stages, not only larvae and nymphs (Figure 1). Our findings further indicate that preferential infestation of *I. ricinus* larvae feeding on deer may directly interfere with the transmission cycles of tick-borne pathogens, such as *A. phagocytophilum*, that utilizes deer as amplification hosts (Table 2).

Here, we detected *I. hookeri* in ticks collected in 18 of the 19 locations around the Netherlands with a prevalence in questing *I. ricinus* nymphs ranging from 0.1% to 16% (Figure 1 and Appendix A). This wasp has a widespread distribution in Europe [27,29,31,33,34,35] and various factors, including microclimate, tick density, and tick–host abundance, were proposed as possible determinants for the high variation in infestation rates [31,40,41]. A prospective study of ten years reported a decrease in *I. scapularis* larvae and nymphs together with a decrease in the *I. hookeri* prevalence in questing nymphs when deer populations sharply declined [40]. Similar observations were made in our cross-sectional study, where the encounter probability of ungulates, most notably roe deer, was positively associated with the prevalence of all the life stages of *I. ricinus* [13] and with the *I. hookeri* prevalence in questing nymphs (Figure 2, left panel). Interestingly, the prevalence was indifferent to the encounter probability of rodents (Figure 2, right panel). Fluctuating rodent populations have been shown to affect density of *I. ricinus* nymphs [18,19]; however, presence of deer is generally responsible for high density of ticks due to their role as propagation hosts [15,17,42].

In addition, deer play a role in attracting female *I. hookeri* over some distance [38,39]. In laboratory tests, it has been demonstrated that *I. hookeri* females preferred mostly unfed *I. ricinus* nymphs for oviposition over unfed larvae, engorged larvae and fully engorged nymphs [27], however, engorging ticks were not included in the experiment. In addition, it is not known how often wasps and questing nymphs are in spatial proximity in the wild. A strategy for female *I. hookeri* is to locate a nymph with help of chemical cues from a vertebrate host [32,43]. In laboratory experiments, *I. hookeri* females were attracted by carbon dioxide, which is an unspecific vertebrate cue, and by odors from roe deer feces, roe deer hair, and wild boar hair. In contrast, they were not attracted to odors derived from field mice, cattle, and rabbits [32]. Locating and parasitizing feeding ticks on their mammal hosts may be advantageous for *I. hookeri* wasps in several ways. First, the density of ticks feeding on a deer is expected to be (much) larger than the density of ticks in the vegetation [44]. Second, the probability of an engorging larva to successfully molt to a questing *I. ricinus* nymph is (much) higher than that of a questing larva [11,45].

Overall, the interaction of the wasp with the propagation host (deer) of its tick host ensures to a large extent the continuation of its lifecycle. This tri-trophic interaction makes it very difficult, if not impossible, to disentangle the effect of deer abundance on the tick and wasp populations, as well as the effect of the wasp on the tick population. Indeed, although the parasitic wasp is killing nymphs upon its emergence, hence interfering with the life cycle of ticks, our study was unable to detect any effect of the occurrence of the wasp on the population sizes (densities) of the different life stages of *I. ricinus* (Figure 1).

In this study, the wasp was detected in questing larvae and nymphs, but not in adult ticks (Table 1). Wasp DNA has previously been detected in questing nymphs and adults of *I. ricinus* with significantly higher prevalence in nymphs [33]. Our results are in line with the idea that nymphs are the main developmental stage affected by the parasitoid wasp. In addition, until now, the emergence of *I. hookeri* has been observed only in fully fed *I. ricinus* nymphs [27,30]. Nevertheless, parasitization itself might already occur at the larval stage of questing ticks to some extent. Although we detected the wasp DNA in two larvae (Table 1), we can imagine that they might have been parasitized while briefly feeding on a vertebrate rather than while questing. Some larvae in the vegetation already fed on a host and inadvertently detached due to grooming. The same explanation is used for finding *B. burgdorferi* s.l. infected questing larvae (Table 1, [46]). The results on co-occurrence of both *I. hookeri* and tick-borne pathogens in questing nymphs are contrary to what we expected (that *I. hookeri* parasitize ticks in the vegetation). Instead, our results support the idea that parasitization takes place when a larva feeds on a vertebrate. We observed a positive association between infestation with *I. hookeri* and infection with *A. phagocytophilum*, which is a pathogen that a *I. ricinus* larva acquires while feeding on a deer, an *A. phagocytophilum*-competent host (Table 2; [47,48,49]). Thus, both acquisition of a bacterium and parasitization by *I. hookeri* occurs simultaneously. Interestingly, we detected a strong negative association of *I. hookeri* with *B. afzelii* and *N. mikurensis*, rodent-associated pathogens (Table 2). The absence of Lyme spirochetes and *Babesia microti* in *I. scapularis* nymphs infested with parasitic wasps has been observed before [50]. These results suggest that wasps rarely parasitize larvae feeding on rodents [51,52] and questing nymphs infected with these pathogens. Our results further imply that *I. hookeri*, which infests feeding *I. ricinus* larva, can survive its molting to the nymphal stage (transstadial transmission). The efficiency of the transstadial transmission of the wasp as well as the effects on the survival on the infested tick is unknown, and warrants further investigation.

One of the questions of this study was whether *I. hookeri* decreases disease risk by reducing tick and/or their associated pathogen communities. Obviously, a fraction of the nymphs will not complete its life cycle, because of predation by the wasp, and therefore the abundance of the wasp has a negative impact on the life cycle of *I. ricinus*. Given that survival of wasps depends on its interaction with deer (the propagation hosts of *I. ricinus*), a diminishing or mitigating effect by the wasp on the tick population is rather unlikely (Figure 1). The parasitic wasp may influence tick-borne pathogen cycles to various extent, as it co-occurs in questing nymphal ticks less frequently with *B. afzelii* and *N. mikurensis* and more frequently with *A. phagocytophilum*. Future studies could investigate the potential long-lasting effect of *I. hookeri* on *A. phagocytophilum* and other deer-associated pathogens, such as *Babesia* spp. Investigation into comprehensive tick-borne disease risk ultimately requires the estimate DIN of these and other tick-borne pathogens.

The results from the artificial tick feeding assay show that molecular detection of *I. hookeri* DNA indicates presence of viable wasps as adult wasps successfully emerged from artificially blood-fed nymphs (Table 3). A total of 7 of the 172 nymphs (4.1%) were killed by emerging wasps (Figure 3), which is in concordance with one previous study [27], but not another [34]. The number of emerged parasitoids from a single *I. ricinus* nymph in our assay ranged from 3 to 5 wasps per tick, which were 1–15 in a German study and 2–20 in a Slovak study [27,34]. About 8% of the nymphs did not molt, and no wasp emerged (Table 3). From these not molted nymphs, 64% was infested with *I. hookeri*, which is comparable to a previous study [27]. The nymphs of that study were dissected one year later and dead *I. hookeri* wasps were found inside [27]. Adult parasitoids may not be able to emerge if they cannot consume all the material inside the nymphs [25]. Interestingly, *I. hookeri* DNA was also detected in 10 of 151 (6.6%) engorged ticks that successfully molted to adults and thus survived infestation with the parasitic wasp. This observation indicates that the development of wasp eggs was suppressed, but the mechanism behind it remains unknown. In other arthropods, it has been shown that some facultative symbionts may enhance survival of their host infested by a parasitic wasp [53,54].

The infection rate of the tick-borne pathogens in the ticks from the feeding assay is not in line with the observations in the field (Table 3). The absence of *B. burgdorferi* s.l. and *N. mikurensis* could be explained by antibiotics in the assay or by their incompatibility with the innate immune components in bovine blood [55]. Admittedly, little is known about antibiotic susceptibility, immune (in) compatibility, and tissue tropism of *N. mikurensis*. In contrast, *A. phagocytophilum* prevalence in *I. ricinus* nymphs that fed in the feeding assay was significantly higher (17.6%) than nymphs from the vegetation (Table 1 and Appendix A). Perhaps, the intracellular location of *A. phagocytophilum* in ticks might prevent killing by gentamycin [56]. Further improvements of the artificial feeding system are necessary to enable studies of the microbial interactions with the parasitic wasp in more detail. Results from the artificial feeding assays indicate that the molecular detection of the wasp is indicative for the presence of viable wasps in ticks.

## 4. Materials and Methods

### 4.1. Tick Collection

*Ixodes ricinus* of all life stages were collected in 2019 by dragging a blanket of 1 m^2^ from two locations: Buunderkamp and Amsterdamse Waterleidingduinen (Appendix A), the Netherlands. Ticks were identified to species level using morphological keys [57]. A proportion of nymphs collected in Amsterdamse Waterleidingduinen was used in an artificial blood-feeding assay (see below). The remainder of nymphs were tested by PCR-based methods for the presence of tick-borne pathogens and *I. hookeri.*

### 4.2. Cross-Sectional Study

Extensive field surveys had been carried out previously in 19 sites located in forested areas in the Netherlands in 2013 and 2014 [13,36]. Data were collected on the density of questing *I. ricinus* (blanket dragging), vertebrate communities (camera and live trapping), and on the infection rates of tick-borne pathogens (qPCR detection). Only data on *A. phagocytophilum*, *N. mikurensis*, and *B. afzelii* were used in this study. Details from these forest sites are provided in Appendix A. All handling procedures of this study were approved by the Animal Experiments Committee of Wageningen University (WUR-2013055 and WUR-2014019) and by the Netherlands Ministry of Economic Affairs (FF/75A/2013/003).

### 4.3. Detection of I. hookeri and Tick-Borne Pathogens

DNA extraction from the individual questing ticks was achieved by alkaline lysis in ammonium hydroxide [58]. The lysates were stored at 4 °C. Samples were tested with qPCRs for presence of *A. phagocytophilum* [47], *B. burgdorferi* s.l. [59], and *N. mikurensis* [60]. For the study described here, the presence of *I. hookeri* DNA was detected by qPCR targeting a 104-bp fragment of the Cytochrome Oxidase I gene using primers 5′-AGA TGT TGA TAC TCG AGC TT-3′, 5′- AAT TTT ATT CCA TTT ATT GAA GCT A-3′ and a probe 5′-ATTO647- TGC TGT TCC AAC AGG AGT AAA AGT TTT TAG ATG A-BHQ2-3′. The specificity of this newly developed qPCR was determined as described previously [31,59]. In short, qPCR-positive samples were subjected to conventional PCR targeting a fragment of the 16S rRNA gene [31]. Both strands of PCR products were sequenced using Sanger Sequencing (Baseclear, Leiden, Netherlands). Resulting sequences were compared with sequences in Genbank using BLAST. A DNA lysate from an *I. hookeri* specimen, which was both morphologically and genetically identified, was used as positive control [31]. All qPCRs were carried out on a Light Cycler 480 (Roche Diagnostics Nederland B.V, Almere, the Netherlands) in a final volume of 20 μL with iQ multiplex Powermix, 3 μL of sample, and 0.2 μM for all primers and different concentrations for probes (30176908). Positive controls and negative water controls were used on every plate tested. To minimize contamination and false-positive samples, the DNA extraction, PCR mix preparation, sample addition, and qPCR analyses were performed in separated air locked dedicated labs.

### 4.4. Co-Infection Analysis

A Fisher’s exact test was applied to explore correlations between tick-borne pathogens and *I. hookeri*. For this, the expected co-occurrence was calculated assuming independent acquisition of tick-borne pathogens and of parasitoid wasps by multiplying their prevalence estimates and the observed density of nymphal ticks [13].

### 4.5. Association of I. hookeri Prevalence in Questing Nymphs with the Density of I. ricinus

We performed regression analyses to investigate associations of *I. hookeri* prevalence in questing *I. ricinus* nymphs with densities of *I. ricinus* larvae (DOL), nymphs (DON), and adults (DOA) of *I. ricinus*. Because *I. hookeri* prevalence is represented as proportional data, we chose a binomial generalized linear model taking into account sample size with the logit link transform. A likelihood ratio test was performed to assess the goodness of fit of all models. The ranges of DOL, DON, DOA, as well as *I. hookeri* prevalence in questing nymphs are provided in Appendix A. The prevalence of *I. hookeri* was calculated based on a subset of samples tested. Model building was performed in R version 3.6.1 “Action of the Toes” [61].

### 4.6. Association of I. hookeri Prevalence with Vertebrate Encounter Probability

In order to find out whether *I. hookeri* displays the observed vertebrate host preference as inferred from the co-occurrence data, we considered a scenario in which numbers of *I. hookeri* presence in the local tick collections follow the beta binomial distribution. Furthermore, the beta mean was considered to relate (via the logit link) to the local ungulate population as measured by encounter probability [13]. The ungulate population consisted of four species: roe deer (*Capreolus capreolus*), fallow deer (*Dama dama*), red deer (*Cervus elaphus*), and wild boar (*Sus scrofa*). The rodent population consisted of two (major) species, namely wood mouse (*Apodemus sylvaticus*) and bank vole (*Myodes glarelolus*). Other rodent species, which occurred to a negligible extent, were: field vole (*Microtus agrestis*), common shrew (*Sorex araneus*), and pygmy shrew (*Sorex minutus*). The model was fitted by maximizing the beta binomial likelihood to the numbers of *I. hookeri* presence and absence per forest site. The likelihood-ratio test was applied to test whether *I*. *hookeri* prevalence is significantly associated with the local ungulate population. In addition to the ungulates, an association of rodents to *I. hookeri* prevalence was tested by following the same procedure. The calculations were performed using R [62].

### 4.7. Artificial Blood-Feeding of I. ricinus Nymphs and Examination of Wasp Emergence

In order to determine the wasp emergence rate, *I. ricinus* nymphs were placed on artificial blood-feeding units which were prepared according to previously described methods [63,64]. Between 50 and 60 ticks were placed in each of the six blood-feeding units. The nymphs were fed on heparinized bovine blood, which was supplemented with glucose (4 g/L), fungizone (2.5 μg/mL), gentamycin (50 μg/mL), and 5 μL of 100 mM ATP solution per 4 mL of blood. Cow blood was obtained from Carus (Wageningen University, The Netherlands) under animal ethics protocol no. AVD1040020173624 and from the Faculty of Veterinary Medicines, Utrecht University.

Blood was replaced every 24 h. Engorged and detached ticks were collected and stored individually in 2 mL Eppendorf tubes with pierced lids, which were kept in a desiccator, with approximately 90% relative humidity at room temperature and observed daily for parasitic wasp emergence. After the blood-feeding experiment, all engorged ticks were also tested for presence of tick-borne pathogens and *I. hookeri* with PCR-based methods.

## 5. Conclusions

Ungulates, particularly deer, are important drivers of tick populations and facilitate the infestation of *I. ricinus* by *I. hookeri*. This double role of deer diminishes the negative effect of the wasp on the tick abundance by killing *I. ricinus*. As this wasp infests *I. ricinus* larvae feeding on deer rather than on rodents, it has no direct interference with the transmission cycle of *B. afzelii*. Taken together, natural *I. hookeri* populations have a minimal impact on TBD risk. Insights in these ecological interactions might provide new food for thoughts on the biological control of *Ixodes ricinus*-borne diseases.

## Figures and Tables

**Figure 1 pathogens-09-00339-f001:**
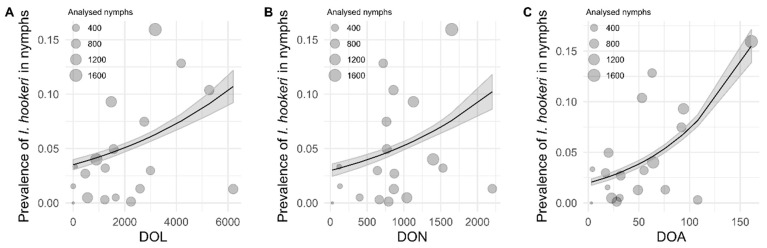
Associations of the prevalence of *I. hookeri* in questing nymphs with (**A**) the density of *I. ricinus* larvae (DOL), (**B**) density of nymphs (DON), and (**C**) density of adults (DOA) in the 19 forest sites (Appendix A). The density is presented per 1200 m^2^. Grey shading around the black regression line represents standard errors. All presented associations are significant (*p* < 0.0001). Equations of all models, AIC values and results of the likelihood ratio test are provided in Appendix A.

**Figure 2 pathogens-09-00339-f002:**
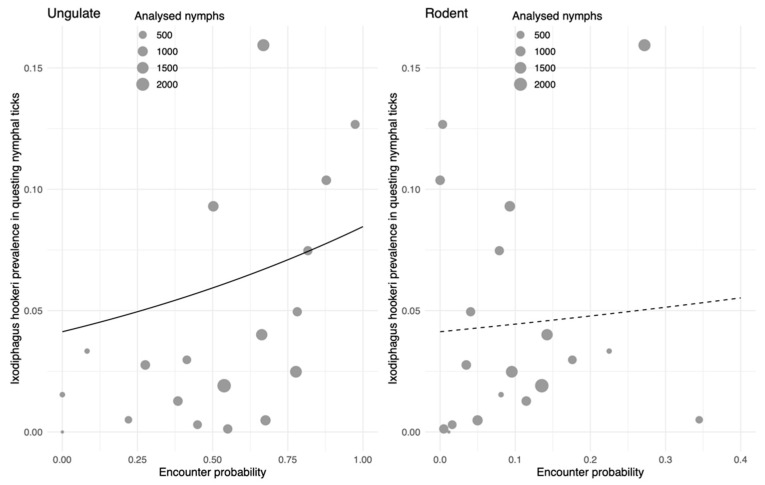
Association of vertebrate species groups (ungulates and rodents) with the occurrence of *I. hookeri* in questing nymphs in the 19 forest sites. The occurrence is presented as the infestation prevalence of *I. hookeri* in questing nymphs (density of infected nymphs; DIN) at each site of 1200 m^2^. Horizontal axis: a function of encounter rates (see methods). A circle represents a single forest site. Best-fit beta binomial model is visualized by a solid line (a significant relationship to the host species, *p* = 0.018) or a dashed line (not significant, *p* = 0.7).

**Figure 3 pathogens-09-00339-f003:**
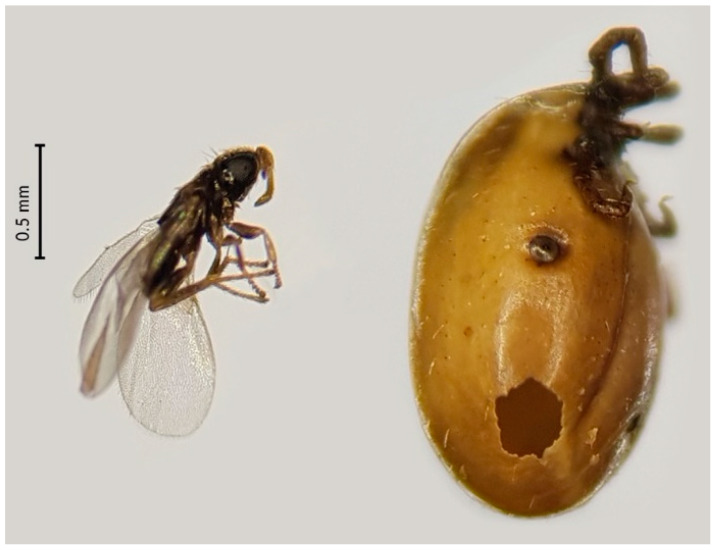
*Ixodiphagus hookeri* (left) and an engorged nymph of *I. ricinus*, which the parasitoid wasp emerged from (right). The field-collected nymph was fed to repletion using an artificial blood-feeding assay. During the molting process, a wasp emerged in the test tube. Photos were taken with a mobile phone and three images were processed in photo editing software to one. Original photos are in Appendix A.

**Table 1 pathogens-09-00339-t001:** The occurrence of tick symbionts in questing *I. ricinus* ticks.

Symbiont	Larvae (*n* = 367)			Nymphs (*n* = 684)			Adults (*n* = 78)		
	*n*	%	(Range)	*n*	%	(Range)	*n*	%	(Range)
*I. hookeri* %	2	0.5%	(0.1–2.5)	30	4.4%	(3.0–6.2)	0	0%	(0.0–4.6)
*B. burgdorferi* s.l. %	1	0.3%	(0.0–1.5)	82	12%	(9.6–14.7)	13	16.7%	(9.2–26.8)
*A. phagocytophilum* %	0	0%	(0.0–1.0)	19	2.8%	(1.7–4.3)	7	9%	(3.7–17.6)
*N. mikurensis* %	0	0%	(0.0–1.0)	30	4.4%	(3.0–6.2)	4	5.1%	(1.4–12.6)

Different life stages of questing *I. ricinus* were collected and tested for the presence of *I. hookeri* and tick-borne pathogens. Occurrence is presented as *n* (number of positive ticks), prevalence (%), and the 95% confidence intervals of the prevalence (range), which is calculated according to Armitage et al. [37].

**Table 2 pathogens-09-00339-t002:** Observed and expected co-occurrence of *I. hookeri* and tick-borne pathogens in *I. ricinus* nymphs.

	*A. phagocytophilum*	*B. afzelii*	*N. mikurensis*
Observed co-occurrence	72	4	9
Expected co-occurrence	26	17	46
Odds ratio	3.3	0.2	0.2
*p*-value	<0.001	<0.001	<0.001

Questing nymphs (*n* = 13,967) from the 19 forest sites were tested for the presence of *I. hookeri*, *A. phagocytophilum*, *B. afzelii*, and *N*. *mikurensis* DNA. Odds ratio >1 and <1 indicates increased and decreased co-occurrence, respectively. A Fisher’s exact test was used to test the statistical significance for an association.

**Table 3 pathogens-09-00339-t003:** The presence of pathogens and *I. hookeri* in nymphs, which successfully blood-fed in the assay.

Ticks		*I. hookeri* %		*A. phagocytophilum* %		*B. burgdorferi* s.l. %		*N. mikurensis* %	
Female	(*n* = 64)	7.8	(2.6–17.3)	15.6	(7.8–26.9)	0.0	(0.0–5.6)	0.0	(0.0–5.6)
Male	(*n* = 87)	5.7	(1.9–12.9)	19.5	(11.8–29.4)	0.0	(0.0–4.2)	0.0	(0.0–4.2)
Not molted	(*n* = 14)	64.3	(35.1–87.2)	14.3	(1.8–42.8)	0.0	(0.0–23.2)	0.0	(0.0–23.2)
With wasps	(*n* = 7)	100	(59.0–100.0)	28.6	(3.7–71.0)	0.0	(0.0–41.0)	0.0	(0.0–41.0)
Total	(*n* = 172)	15.1	(10.1–21.4)	18.0	(12.6–24.6)	0.0	(0.0–2.1)	0.0	(0.0–2.1)

Field-collected, questing nymphs were artificially fed in blood-feeding units in the laboratory, and the emergence of parasitoid wasps was monitored. After that, all ticks were analyzed by molecular methods for the presence of pathogens and *I. hookeri*. Between brackets are the 95% confidence intervals, as calculated according to Armitage et al. [37].

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
