# Peer review of "Tripartite Interactions among Ixodiphagus hookeri, Ixodes ricinus and Deer: Differential Interference with Transmission Cycles of Tick-Borne Pathogens"

_pathogens, 2020, doi:10.3390/pathogens9050339_

Round 1

Reviewer 1 Report

The authors describe an interesting relationship between a parasitic wasp, a tick parasite and vertebrate host. With careful editing and consideration I think this manuscript will be a valuable addition to the field.

Please see attached file for additional reviewer comments and editing suggestions.

Reviewer 2 Report

The authors investigate the role that Ixodiphagus hookeri may have in mitigating tick-borne diseases.  The authors conclude that the presense of I. hookeri  may interfere with the transmission cycle of Anaplasma phagocytophilum but likely has no impact on two other Ixodes ricinus-borne pathogens.  They propose some interaction between the former, given that a larger proportion of ticks (nymphs) collected from deer were infected with I. hookeri. This conclusion is made because higher infection with I. hookeri was positively correlated with the encounter probability of ungulates and the density of I. ricinus.

specific comments:

  • Table 1 and 3, and lines 246 and 254 - all commas should be changed to decimal points, i.e. (0,1-2,5) should be (0.1-2.5)
  • Line 117 - this is the first time these acronyms are used in the manuscript.  Please define DOL, DON ad DOA here.
  • Lines 142-153 - could you add an additional table or maybe more columns to table 3 showing co-infections of these ticks with I. hookeri and the tick-borne pathogens. Were co-infected nymphs more or less likely to produce live wasps, etc.?
  • Line 176-177 - Please elaborate on this, I am not sure I understand what you are trying to say and what your basis for concluding that I. hookeri may interfere with the transmission cycle of A. phagocytophilum.  Your co-infection data shows that more nymphs that expected are co-infected with I. hookeri and A. phagocytophilum. How does this data lead to that conclusion?  Is A. phagocytophilum only associated with deer in the Netherlands? In USA, A. phagocytophilum is associated with several species of rodents.
  • Lines 259-266 - could you discuss why antibiotics in the blood may have decreased detection of B. burgdorferi and N. mikyrensis, but not A. phagocytophilum.

Reviewer 3 Report

Dear Author,

Submitted manuscript - Tripartite interactions among Ixodiphagus hookeri, Ixodes ricinus and deer: differential interference with transmission cycles of tick-borne pathogens, is a well-documented and appropriately written draft for the molecular detection and ecological relationship among Ixodiphagus hookeri wasp parasitizing on Ixodes ricinus tick’s nymph and their horizontal transmission through deer Capreolus capreolus in Netherland forest area.

Here are some minor comments, if author will incorporate and taken into consideration, manuscript become more convincing to the readers.

Throughout the manuscript, there are two species one tick parasitized wasp and second TBP vector species harboring on mammals, more prevalently been used along with their life stages. It has been reflected somewhere (Ex. Line no. 64) that, sentences are looking incomplete due to not mentioning of the species name while talking about their life stages. Hence, it is suggested to take care of that ambiguity to avoid any confusion to the reader in whole manuscript.

Figure S2: Forest sites in which ticks were sampled. As the GIS dimensions of these forest sites are not being mentioned, hence in an easy way it is suggested to add in the inset of mainland area from where this forest location was derived. Also, if it is possible, the supplementary photographs, map and pictures, submit it in a different file format, not in excel.

Mention the platform of statistical software been used in the study for data analysis in the methodology section. As well as the consent statement of Institutional animal ethical committee.

Check the spelling, English grammar and formatting one more time at your end.

Even though, I appreciate the efforts made by authors/acknowledging staffs to work on field and at laboratory level and derive the finding in contextual manner.

All the Best.

Round 2

Reviewer 1 Report

ok